Nest-building behavior of Monk Parakeets and insights into potential mechanisms for reducing damage to utility poles

Burgio Kevin R. kevin.burgio@uconn.edu
Rubega Margaret A.
Sustaita Diego 1
Department of Ecology and Evolutionary Biology, University of Connecticut , Storrs, CT , USA
Kramer Donald
1 Current affiliation: Department of Ecology & Evolutionary Biology, Brown University, Providence, RI, USA.

Electronic publication date: 2014 Sep 30
Publication date: 2014
Volume: 2
Electronic Location ID: e601
Received 2014 Jul 25; Accepted 2014 Sep 9
Copyright: © 2014 Burgio et al.
Copyright year: 2014
Copyright holder: Burgio et al.
License: This is an open access article distributed under the terms of the Creative Commons Attribution License, which permits unrestricted use, distribution, reproduction and adaptation in any medium and for any purpose provided that it is properly attributed. For attribution, the original author(s), title, publication source (PeerJ) and either DOI or URL of the article must be cited.
License URL: https://creativecommons.org/licenses/by/4.0/

Keywords: Monk Parakeet, Myiopsitta monachus, Nesting behavior, Management, Human/wildlife interactions, Invasive species

Funding: National Science Foundation # DEB-0823772 # DGE-0753455 University of Connecticut Office of Undergraduate Research Treibick Family Foundation Katie Bu Memorial Scholarship Fund Our research was supported by funding provided through an REU supplement to National Science Foundation grant (# DEB-0823772) to M Rubega and J Silander, and awards to KR Burgio from the National Science Foundation (# DGE-0753455) the University of Connecticut Office of Undergraduate Research, the Treibick Family Foundation, and the Katie Bu Memorial Scholarship Fund. The funders had no role in study design, data collection and analysis, decision to publish, or preparation of the manuscript.

==============================
The Monk Parakeet (Myiopsitta monachus) commonly uses utility poles as a substrate for building large, bulky nests. These nests often cause fires and electric power outages, creating public safety risks and increasing liability and maintenance costs for electric companies. Previous research has focused on lethal methods and chemical contraception to prevent nesting on utility poles and electrical substations. However, implementation of lethal methods has led to public protests and lawsuits, while chemical contraception may affect other than the targeted species, and must be continually reapplied for effectiveness. One non-lethal alternative, nest removal, is costly and may not be a sustainable measure if Monk Parakeet populations continue to grow. In order to identify cost-effective non-lethal solutions to problems caused by Monk Parakeet nesting, we studied their behavior as they built nests on utility poles. Monk Parakeets initiate nests by attaching sticks at the intersection of the pole and electric lines. We found that parakeets use the electric lines exclusively to gain access to the intersection of lines and pole during nest initiation, and continue to use the lines intensively throughout construction. Monk Parakeets also have more difficulty attaching sticks during the early stages of nest construction than when the nest is nearing completion. These findings suggest that intervention during the earlier stages of nest building, by excluding Monk Parakeets from electric lines adjacent to poles, may be an effective, non-lethal method of reducing or eliminating parakeets nesting on, and damaging, utility poles.

The economic damage and the management of non-native species costs about 120 billion dollars per year in the United States alone (Pimentel, Zuniga & Morrison, 2005). One such species, the Monk Parakeet (Myiopsitta monachus), a bird native to South America, has invaded numerous continents over the past 50 years (e.g., Roll, Dayan & Simberloff, 2008; Strubbe & Matthysen, 2009). Despite their globally demonstrated ability to become naturalized in new areas, Monk Parakeets remain popular in the pet trade and presently account for 97% of reported parrot exports from South America (Bush, Baker & MacDonald, 2014). In the recent past, Monk Parakeets have shifted from building their large, bulky stick nests exclusively in trees (Eberhard, 1996) to also building on utility poles and substations, which frequently cause fires and electrical power outages (Avery et al., 2006; Reed et al., 2014).

In order to reduce the impact of Monk Parakeet nesting, US electric companies have been conducting nest removals and other management actions, which costs them millions of dollars each year (Avery, Yoder & Tillman, 2007). Aggressive management practices, including trapping and euthanasia, have led to protests and lawsuits by animal rights activists (Russello, Avery & Wright, 2008; Friends of Animals v. United Illuminating 124 Conn. App. 823, 2010). The practice of routinely removing nests from utility poles may not be cost effective, considering that one pair of parakeets can rebuild a nest in less than two weeks (Avery et al., 2002). Because of the legal, public relations, and financial difficulties electric companies face in dealing with expanding Monk Parakeet populations, finding a solution that is cost-effective, safe, and non-lethal is critical for reducing conflict over the control of Monk Parakeets nesting on utility poles.

We investigated Monk Parakeet nest initiation and building behavior to identify possible mechanical (in contrast to lethal) solutions. We report here a study aimed at answering the following questions: (1) Is there a critical access point that Monk Parakeets use in order to build their nests on the utility poles? Based on preliminary observations, we hypothesized that Monk Parakeets primarily use electric distribution lines (henceforth “electric lines”), as opposed to the pole or transformers, to gain access to the nest site throughout the nest-building process. (2) If there is a critical access point, does it change as nest construction progresses? We hypothesized that Monk Parakeets use only electric lines early in construction and start using other structures, especially the nest itself, as landing platforms as the nest gets bigger. (3) Is there a point during which nest construction is easiest? We hypothesized the presence of existing nest structure (sticks) makes it progressively easier to attach additional sticks. To answer these questions and test our hypotheses, we quantified the behavior of Monk Parakeets as they transferred sticks from trees to the nest site.

Methods

Observations

From April to December 2009, during which United Illuminating (UI) personnel removed Monk Parakeet nests from 69 electric utility poles in Stratford, West Haven, and Hamden, Connecticut, we systematically monitored each pole beginning the day after nest removal. At every pole where we found Monk Parakeets actively rebuilding, we conducted a standardized 30-min observation protocol. We sampled opportunistically, collecting data on 18 different days, between 8:40 and 16:15, with the majority of observations conducted before noon. Our research methods were examined, and classified as exempt from further review by the University of Connecticut’s Institutional Animal Care and Use Committee (Exemption # E08-006) because our work was purely observational.

We recorded the total number of landings on the utility pole or surrounding structures attached to the pole, including electric lines, transformers, and nests (Fig. 1) by Monk Parakeets carrying sticks (Fig. 2). We also recorded the location of the landing immediately preceding stick placement on the nest and scored each stick associated with a landing as either attached or dropped. We classified the stage of building by relative nest size, using transformers as a size reference. Transformers in our study area averaged 0.77 m tall and 0.48 m in diameter, with little variation among the four most common types (H Loso, 2014, unpublished data). We classified nests less than 25% of the size of the transformer on the pole as “Beginning”; 25%–50% as “Small”; 50%–75% as “Medium”; and greater than 75% as “Large”.

Figure 1 Beginnings of a Monk Parakeet nest on a utility pole.

Utility pole (A) with the beginnings of a Monk Parakeet nest (B). During construction, the first sticks are typically placed between the utility pole (A), transformer (C), and electric lines (D). Because birds carry sticks in their beak, they cannot climb down to the nest initiation site from either the pole, or the transformer. Photo: KR Burgio.

Figure 2 Monk Parakeet bringing a stick to the nest site.

The partially completed nest structure is to the left of the bird. Note the method of stick carrying; parakeets hold one end of the stick in their beak, letting the rest of the stick trail, rather than carrying the stick crosswise. We never saw a Monk Parakeet carry sticks any other way. The parakeet will walk along the electric lines to the nest structure, and add this stick to the nest. Photo by KR Burgio.

Statistical analyses

Individual parakeets were not uniquely identifiable, and the number of birds present during construction ranged from one to five (or more). Furthermore, parakeets spent variable amounts of time across poles during nest building, resulting in variable numbers of spatially and temporally non-independent observations (3–48), of variable durations, across the 19 utility poles we sampled. Therefore, we employed non-parametric randomization tests and generalized linear mixed-models (described below) in an effort to account for some of the lack of independence of observations among individual parakeets and nest poles.

We began by testing whether Monk Parakeets landed on electric lines prior to attaching a stick to their nest more often than they landed on utility pole structures. For this analysis we treated each pole as the unit of replication, and therefore randomly selected one observed landing from each of the 19 poles. We then subjected this subset of line, nest, and pole observations to a goodness of fit (G-test) to an extrinsic hypothesis of equal usage (i.e., 1:1:1 ratio of landings across structures) in Excel (Version 2003; Microsoft Corporation, Redmond, WA), based on formulae from Sokal & Rohlf (1995). We then repeated these two steps 5,000 times using a custom-built Excel macro, and determined the proportion of tests that were significantly different from random (α = 0.05).

To model the probability that Monk Parakeets land on the electric lines as a function of nest size, we conducted a generalized linear mixed model (Bolker et al., 2009) with “pole” specified as a random grouping factor, in R (R Development Core Team, 2009) using glmmML (Göran, 2013). Because there were too few instances of landings on the poles to warrant the added complexity of a multinomial model, we treated the response as binary (landing on lines = 1, landing elsewhere = 0), assuming a binomial distribution and a logit link function. Although we initially assigned each nest size an ordinal value, we felt it was appropriate to treat nest size as a continuous variable under the tacit assumption that the size intervals were roughly consistent between classes, and the observed linear relationship between nest size and the raw proportions of landings on the electric lines. Nevertheless, we performed a series of alternative analyses (not shown) with nest size specified as a multi-level categorical variable, and the results were consistent. We used this same analytical approach to model the probability of stick attachment success (1 = attached, 0 = dropped) as a function of nest size. We assessed adequacy of the models according to procedures described by Rhodes et al. (2009).

Results

We observed a total of 405 landings across 19 nest poles. Monk Parakeets carried sticks in their beak (Fig. 2); we never saw an instance of a parakeet transporting a stick by any other method, and never observed parakeets transfer sticks from beak to footholds. Monk Parakeets landed exclusively on the electric lines during nest initiation and early construction (Figs. 3 and 4). Further, electric lines were landed on more frequently than any other location throughout all stages of nest-building. Across all observations, including later stages of nest building, 88.7% of landings were on electric lines, and the use of electric lines was significantly greater than expected (8.08 ≤ G2 (Williams’-corrected) ≤40.33; 8.60 × 10−10 ≤ P ≤ 0.015; Fig. 3) in all 5,000 tests of resampled data sets. Although landings on the lines still predominated, as their nests grew larger parakeets progressively landed either on the pole or the nest itself (parameter estimate = −1.13 ± 0.32, z = −3.50, P = 0.0046, df = 402, residual deviance = 209.1; Fig. 4). Although generally successful at placing sticks, parakeets dropped sticks more frequently when nests were small and were progressively more successful at actually attaching sticks as nests grew larger (parameter estimate = 0.398 ± 0.159, z = 2.50, P = 0.0124, df = 402, residual deviance = 326.5; Fig. 5). Monk Parakeets in our study never picked up a stick once dropped (n = 57 of 405 observations; 14%); parakeets always flew back to cut off a fresh branch instead.

Figure 3 Bar graph of mean simulated frequencies of Monk Parakeet landing site selection prior to accessing the nest during construction.

Error bars represent mean (± SD) frequencies of 5,000 random samples of landing observations across 19 utility poles, for each of three potential landing positions. Expected frequency line shows where mean frequencies of landing on any site would have been expected if parakeets were landing equally on all landing sites available. All resampled data sets showed significantly more frequent landing on electric lines than the null expectation (all P ≤ 0.015).

Figure 4 Bar graph of the use of electric lines as an access point for Monk Parakeets during nest construction.

Bars represent the proportion of observed landings of parakeets carrying sticks to the nest-site as nests became larger for each landing position, either electric lines or “other” (nests or poles) (n = 405 observations over 19 nests). Numbers above the bars are the raw frequencies for each category. Monk Parakeets used electrical lines exclusively during nest initiation, and the proportions of landings on the electric lines decreased with increasing nest size.

Figure 5 Bar graph of stick attachment success/failure as a function of nest size for nesting Monk Parakeets.

Bars represent the proportion of the number of sticks successfully attached or dropped as nests became larger (n = 405 observations over 19 nests). Numbers above the bars represent the raw frequencies for each category. Stick attachment success increased with increasing nest size.

Discussion

Our data demonstrate that electrical lines are the most frequently used access point for Monk Parakeets building nests on utility poles, and therefore a prime target for non-lethal methods of excluding or interrupting nest building. When starting a nest, the parakeets used the electric lines exclusively. Since they carry sticks only with their beak (Fig. 2), the parakeets require a route to the nest-construction site that they can walk on; climbing would require beak-holds, thus could not be used to initiate a new nest, or to add sticks to an existing construction. Even with access via the lines, the parakeets encountered greater difficulty in attaching sticks when they were initiating a nest than in later stages of nest-building, as demonstrated by the higher frequency of stick dropping. The reduction in stick dropping as nests grew suggests that nest initiation is the stage of nest-building most vulnerable to human intervention in the nest building cycle. Taking the importance of access via the lines and increasing success at stick attachment as building proceeds, we suggest mechanically excluding access to the section of electric lines nearest the pole before the parakeets start building may be both necessary, and sufficient, to keep Monk Parakeets from nesting on utility poles.

Excluding access to the electric lines adjacent to the utility poles sidesteps the need to provide a tight “seal” of structural modifications of the pole, transformer, or attachments previously attempted to prevent stick attachment (Newman et al., 2008). Avery et al. (2006) point out that there are so many configurations of attachments that comprehensive modification is impractical since eliminating all gaps large enough start a nest would require a new design for each individual pole. However, since all utility poles have electric lines in common, excluding Monk Parakeets from these lines would not require a unique solution for each pole.

It is unlikely that preventing Monk Parakeets from nesting on utility poles will provide a means of population control. Before the advent of utility poles in their native range, parakeets nested in trees, and presumably they would nest-build in trees if excluded from poles. There are few data on reproductive success in naturalized populations, and none that would allow comparisons of relative reproductive success between nests in trees and nests on utility poles. Monk Parakeets are considered crop pests in their native range (Bucher, 1992). Thus, population control has been an additional goal of other methods, e.g., chemical contraception and euthanasia. However, Monk Parakeets have yet to cause serious crop damage in the U.S. (Avery, Yoder & Tillman, 2007). Lethal control (specifically shooting) has not effectively reduced populations or suppressed population growth in their native range (Spreyer & Bucher, 1998). Chemical contraception has yet to be deployed at sufficiently large scales to gauge its utility (Avery, Yoder & Tillman, 2007), and carries the added potential risk of affecting non-target native species. Given the appeal of Monk Parakeets to the general public, and the strength of opposition from animal welfare groups, lethal methods will likely face sustained opposition if attempts are made to implement them on a scale likely to have any lasting effect. If naturalized populations of Monk Parakeets continue to grow, research on their population dynamics and range expansion will become more important (e.g., Davis, Malas & Minor, 2014).

Thus far, there is no evidence that the naturalization of Monk Parakeets has any negative ecological impacts in North America (Burger & Gochfeld, 2009). However, little direct research to quantify their ecological effect has been conducted. Unlike cavity nesters (e.g., European Starlings (Sturnus vulgaris)), they do not compete for existing nest sites commonly used by other birds; they are not predatory, and appear not to compete directly for food with native birds. However, a recent study found that non-native Rose-ringed Parakeets (Psittacula krameri) negatively impact foraging in native birds in the United Kingdom (Peck et al., 2014).

Non-native Monk Parakeet populations seem likely to continue growing and expanding their distribution (Russello, Avery & Wright, 2008). Our findings are encouraging in their implications for a potential non-lethal, cost-effective solution to preventing Monk Parakeets from building nests on utility poles. However, more research on this approach is needed. Also, this goal would be furthered, and an understanding of their potential for future impact on native species and crops would be gained, by determining whether there is differential reproductive success among naturalized populations, and by identifying the factors that have limited their range expansion thus far.

Supplemental Information

Supplemental Information 1 Utility Pole Nesting Behavior of Monk Parakeets—Raw Data

Utility Pole Nesting Behavior of Monk Parakeets—Raw Data.

Click here for additional data file.

We thank the United Illuminating Company for their cooperation and access to records. We also thank two anonymous reviewers, C Elphick, C Field, C Cizauskas, and E Jockusch for helpful comments; and J Leibrecht and members of the University of Connecticut Ornithology Research Group for assistance in the lab and field.

Additional Information and Declarations

Competing Interests

Author Contributions

Animal Ethics

The authors declare there are no competing interests.

Kevin R. Burgio conceived and designed the experiments, performed the experiments, contributed reagents/materials/analysis tools, wrote the paper, prepared figures and/or tables, reviewed drafts of the paper.

Margaret A. Rubega conceived and designed the experiments, contributed reagents/materials/analysis tools, reviewed drafts of the paper.

Diego Sustaita analyzed the data, wrote the paper, prepared figures and/or tables, reviewed drafts of the paper.

The following information was supplied relating to ethical approvals (i.e., approving body and any reference numbers):

University of Connecticut Institutional Animal Care and Use Committee (IACUC), Exemption number: E08-006.

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
