# Peer review of "Nest-building behavior of Monk Parakeets and insights into potential mechanisms for reducing damage to utility poles"

_PeerJ, doi:10.7717/peerj.601_

## Round 0.1 · original submission · Minor Revisions

· Academic Editor

Minor Revisions

Both reviewers consider that your study meets the criteria for publication in PeerJ. Reviewer 1 considers that only minor revision is necessary and makes a number of valuable suggestions for improvements in the presentation. This reviewer also suggests an alternative approach to the analysis for you to consider, although this reviewer does not indicate any specific problem with the analysis you have carried out. Reviewer 2 suggests major revisions, primarily shortening the manuscript but also elaborating a bit on what could be done practically to take practical advantage of your findings.

On the basis of these reviews and my own reading of the manuscript, I feel that minor revisions, including both shortening and clarification as well as addressing some considerations of the logic of the study, would make the manuscript suitable for publication.

When you have revised the manuscript, please provide a specific response to each of the suggestions of the reviewers and editors, indicating the change made or the reason for not making a change.

Sincerely,

Donald L. Kramer

Suggestions from the editor

Title: your title is currently misleading because the importance of nest initiation mechanics for reducing damage is only addressed briefly and speculatively; it may have been the motivation, but it was not the topic examined.

Abstract: L9 refers to ‘this problem’ before the problem has been explicitly defined. Clarify.

Introduction:
The Introduction currently uses nearly 3 pages to make a point that could be easily made in 1 page. The problem is not logically well developed, resulting in a high level of redundancy. For example, geographical distribution is referred to in two paragraphs (L4, L16ff), the cost of damage is referred to in three paragraphs (L5ff, L32-33, L41ff). These and related problems should be readily solved by rewriting usng a tight outline based on the logical development of your study question.

L56. ‘an eye toward’ seems a bit colloquial for formal writing and might confuse non-native English speakers.

Note also that you never mention whether a single bird, a pair, or a communal nesting group builds the nest. This may be relevant to the potential number of individuals observed.

Methods:
Scaling nest size to transformer size seems reasonable as a field method. Why not give the dimensions of the transformer here so that readers have a measure in standard units?

L90. ‘exempted’ may not be clear to all readers. I presume this means that the animal care committee decided it did not to evaluate your procedures because it was an observational field study. A slight clarification might help.

L107. Note that you don’t provide any justification for the null expectation of equal probability of landing on the three types of substrate. For a study of selection or choice (e.g. food, habitat, substrate, mate type), the analysis requires examining the relationship between use and availability. If birds were landing ‘at random’, the probability would be presumably based on the availability of each substrate (linear? areal?). It seems obvious that as the nest increases in size, even a random parrot would be more likely to land on the nest. Thus, you could simply refer to a null hypothesis of equal use. However, you really don’t need to show that there is a difference from chance or equality, but only that the frequency of landing on wires is more than the other two categories. I am not suggesting that you engage in an even more complex statistical analysis for this very simple and straightforward point, but you might consider clarifying the null hypothesis in Methods, Results and Captions and addressing in the Discussion the relative amount of ‘wire’ and ‘pole’ available and the increasing size of the nest as considerations when you discuss the preference for landing on wires.

L120-121. My intuition is that when you divide a continuous variable into discrete categories, that you need to consider them as a categorical variable in the analysis. My intuition may be wrong, but it might be useful to confirm the validity of your approach with a statistician.

Discussion
L165. I disagree with your assertion that you have shown electrical lines to be ‘a critical access point’ (also note the grammatical mistake in referring to ‘a . . . points’). You have shown that wires are the most frequently used access point, but ‘critical’ seems to imply that it is an obligate landing site. You have no evidence that without landing on wires, the birds would not be able to initiate a nest. (By analogy, a researcher showing that a particular prey represents 90% of the diet of a predator species could not conclude that this prey is ‘critical’ without examining what the species eats when the particular prey is absent.)

L177. Euthanasia refers to putting to death painlessly to avoid further suffering. I don’t think that this is the appropriate term for shooting. ‘Lethal control’ might be a better alternative expression.

L193 ‘and elsewhere to stay’ is an awkward expression; revise, please.

Figures and captions
Fig. 2. I suggest that the figure shows the frequency rather than importance of landing on each surface type. Inferring ‘importance’ requires some additional assumptions. With reference to the comments on the null hypothesis above, the line represents the expectation for equal rather than random use of each category. Finally, if I understand the statistics correctly, landing on wires is significantly more frequent than expected by the null hypothesis of equal landing, which is not what the caption states. I did not see that you tested the frequency of each category compared to the others, even though this may have been a more appropriate comparison.

Fig. 3 and 4 are not designed to most effectively express the point you wish to make. You are interested in the proportion of landings on wires (Fig. 3) and the proportion of sticks successfully attached (Fig. 4). However, by showing absolute frequencies, the patterns are confounded by variation in the number of observations for each nest stage. Transforming the figures to proportions with number of observations shown above the bars would be more effective. Also note that the first sentence of the caption for Fig. 3 is not correct, both in the use of ‘importance’ instead of frequency (or proportion, if you change the figure as suggested above) and in that it refers to only one part of the graph. Defining only nest stages 1 and 4 may confuse some readers, especially as the definitions do not correspond to the criteria for nest stage provided in the Methods. It is not necessary to repeat the statistics in the captions. In fact, they are more clearly and complete presented in the Results. The statistical information in the caption to Fig. 4 has an unfinished comparison that needs to be completed if it is not removed.

Reviewer 1 ·

Basic reporting

This article meets the Basic Reporting aspect. The authors submitted a well-written manuscript that adequately demonstrates how their research fits into the broader field of knowledge. There are a few grammatical and mechanical issues needing attention. For example: Line 155 "electrical lines are a critical access points"; Line 90 is a one-sentence paragraph; and Line 192 "It seems likely that..." reads like a natural paragraph break. Figures are relevant to the manuscript's content.

Experimental design

This manuscript describes original, unique research on the topic of monk parakeet nesting behavior on electric utility structures. The authors' research greatly contributes to improving our understanding of this nesting behavior. The authors clearly defined their research questions, which are relevant and meaningful. Their research methods were ethical.

The Methods section needs some clarification to support replication by future researchers. For example:
-Define "utility poles", e.g., with or without equipment (capacitor banks, transformers, etc.); electric equipment only, or also telephone and/or cable lines; Do monk parakeets land vertically on utility poles, or on the horizontal crossbeams?
-Define "electric lines". Are they only horizontal (parallel with ground) or are some at various angles and loops?
-What time of day were observations made? Were observations made at the same time each day or randomly sampled at various times?
-How many observation days?
-Provide estimated average size of transformers for clear size-reference points

Statistical Analysis
To account for lack of independence and to simplify the statistical analysis, authors might consider conducting a GLM mixed model and denote poles at random variable.

Validity of the findings

Based on the statistical analyses employed, the authors' state valid conclusions. They answered their three research questions and supported them with appropriate results and figures. It will be interesting to see if results are similar if the authors choose to conduct the alternative statistical analyses suggested above in Experimental Design.

Additional comments

Some suggestions to improve overall clarity of the manuscript:

-Change "birds" to "parakeets"
-Clarify "utility" and "power" as electrical
-Consider removing the unnecessary "thats"

Abstract
Line 2: Clarify "structures" as nests
Line 4: Clarify "solutions" as those to prevent nesting on electric utility substrates
Line 8: Define "populations" as monk parakeet
Line 17: Is the recommendation to reduce and eliminate all nesting bird species, or just parakeets?

Introduction
Define utility companies, as this is a general term applied to electricity, gas, cable, etc.
Line 12: Are there some citations for protests and lawsuits?
Line 20: Introduced monk parakeet populations are established primarily in urban and suburban, not wild, environments.
Line 24: Add "Historically" preceding "They build their nests in trees..."
Line 48: Change "birds" to "nests"?
Line 49: Change "United Illuminating" to "UL" as presented in previous paragraph
Line 65: To be parallel, change "more difficult" to "easier" or change hypothesis to reflect when nesting building is most difficult.
Line 68: Change "building material" to "sticks"?

Methods
Lines 71-74: Consider rewriting for reading ease. For example: "From April to December 2009, during which UI personnel removed Monk Parakeet nests from electric utility poles (n = 19?) in Stratford, West Haven, and Hamden, Connecticut, we systematically monitored each pole beginning the day after nest removal."
Line 74: Consider omitting "During this time frame,"
Lines 73 and 76: Line 73 mentions beginning the day after nest clearing and Line 75 mentions the day of nest clearing; change to be consistent
Line 77: Change "building" to "rebuilding"?
Line 77: Clarify if authors designed/developed the standardized 30-minute observation or followed a previous method (citation)
Lines 79, 143, 144, Fig. 1: Are you using electric lines and wires interchangeably?

Results
Line 152: Is 57 dropped sticks a portion of the 405 landings? Meaning monk parakeets dropped nest sticks 14% of the time?

Discussion
Line 191: Are there any citations for competition at bird feeders?

Reviewer 2 ·

Basic reporting

No comment.

Experimental design

No comment.

Validity of the findings

No comment.

Additional comments

The title of the paper implies that you will indicate how the nest-initiation mechanics could reduced the damage to utilities. In my opinion, you don't do that. You say that utility companies could do something to reduce perching on wires by the birds, but what could they do. What could companies do other than bury the lines? Similarly, if you found that the birds perched on the structure, and not the wire, would your conclusions have been the same? I mean, the birds have to perch somewhere. As long as you have a structure and wires, there is going to be places to perch.
Given the rather straight forward and simple result, that birds perch on wires while building their nest, I find this paper very long. I suggest reducing the length at least by 30%.

---

## Round 0.2 · Minor Revisions

· Academic Editor

Minor Revisions

Sorry for the delayed decision. Your manuscript arrived just as I started moving to a new apartment, and I wasn't able to get to it until my life re-stabilized. Your changes successfully address all 3 sets of comments. However, I noted a few minor errors and typos and some places where wording could be clarified by small changes in the revised manuscript. I made these suggestions using track changes on your track change manuscript. I will send this file to you via the journal. It shouldn't take you long to decide which of these suggestions improve your manuscript. Please note that I will not be available from Sept 13 - 18. PeerJ staff are investigating your query regarding citation of a legal document.

---

## Round 0.3 · accepted · Accept

· Academic Editor

Accept

Thank you for the prompt revision.